epigenetic silencing; No-go RNA decay; RNA-directed DNA methylation; RNA quality control; small RNA.

**Corresponding author:**
Jungnam Cho;
Email: jungnam.cho@durham.ac.uk

S.M. and Y.J.K. authors contributed equally.

**Associate Editor:**
Dr. Yuan Wang

# Small RNA, big defence: Early epigenetic responses to genetic invasion

Seunghui Mun[1], Yang Jae Kang[2] and Jungnam Cho[1]

[1]Durham University, UK; [2]Gyeongsang National University, Republic of Korea

## Abstract

Plants are under constant genetic siege. From viruses and bacteria to transposable elements within their genomes, cells must contend with foreign genetic material. Besides these natural threats, modern biotechnology adds complexity by introducing transgenes to plants. While the integration of such DNA can enhance genetic diversity and confer desirable traits, its foreign origin is typically recognised by the plant cell as a signal of invasion and therefore targeted by the repressive mechanisms. Epigenetic silencing is a central strategy and involves the methylation of DNA and histones. A critical trigger of this silencing is the generation of small interfering RNAs (siRNAs). Although the role of siRNAs in maintaining epigenetic silencing is well established, the initial steps that lead to their production remain incompletely understood. This review discusses the key discoveries on how plant cells recognise foreign nucleic acids and initiate epigenetic silencing, contributing to our broader understanding of genome integrity and defence.

## 1. Introduction

Epigenetic silencing in plants has been primarily studied in repetitive sequences and transposable elements (TEs). RNA-directed DNA methylation (RdDM) is a key epigenetic regulatory mechanism in plants that reinforces and maintains the epigenetic silencing (Gallego-Bartolomé et al., 2019; Matzke et al., 2015; Matzke & Mosher, 2014; Xie et al., 2024). The RdDM pathway is initiated by RNA Polymerase IV (Pol IV), which generates single-stranded RNAs that are converted into double-stranded RNAs by RNA-DEPENDENT RNA POLYMERASE 2 (RDR2). These are then processed into 24-nucleotide (nt) small interfering RNAs (siRNAs) by DICER-LIKE 3 (DCL3) and loaded onto ARGONAUTE 4 (AGO4). In conjunction with transcripts produced by RNA polymerase V (Pol V), 24-nt siRNAs guide the methylation machinery to specific genomic locations. The *de novo* DNA methyltransferase DOMAINS REARRANGED METHYLTRANSFERASE 2 (DRM2) is then recruited to catalyse methylation at cytosines in all sequence contexts (CG, CHG and CHH; H refers to A, T or C).

RdDM must accurately distinguish its genomic targets from essential genes to avoid silencing those critical for plant survival, and this discrimination is mediated by specialised proteins that read epigenetic marks written in the chromatins. For example, Pol IV recruitment to target DNA is coordinated by CLASSY1–4 (CLSY1–4) chromatin remodelers and the SAWADEE HOMEODOMAIN HOMOLOG 1 (SHH1) protein, a histone methylation reader. CLASSY proteins act in a tissue- and locus-specific manner to remodel chromatin and facilitate Pol IV access (Zhou et al., 2018), while SHH1 specifically recognises histone H3 lysine 9 dimethylation (H3K9me2) marks, enriched in repetitive sequences and TEs, and helps stabilise Pol IV binding at these chromatin sites (Law et al., 2011, 2013). In parallel, SU(VAR)3-9 HOMOLOG 2 (SUVH2) and SUVH9, which contain SRA domains that specifically bind methylated DNA, contribute to targeting of Pol V to pre-methylated regions (Johnson et al., 2008, 2014; Liu et al., 2014). This preferential recruitment of Pol IV and Pol V to already methylated DNA and histones highlights the RdDM pathway's primary role in sustaining and reinforcing pre-established silencing marks.

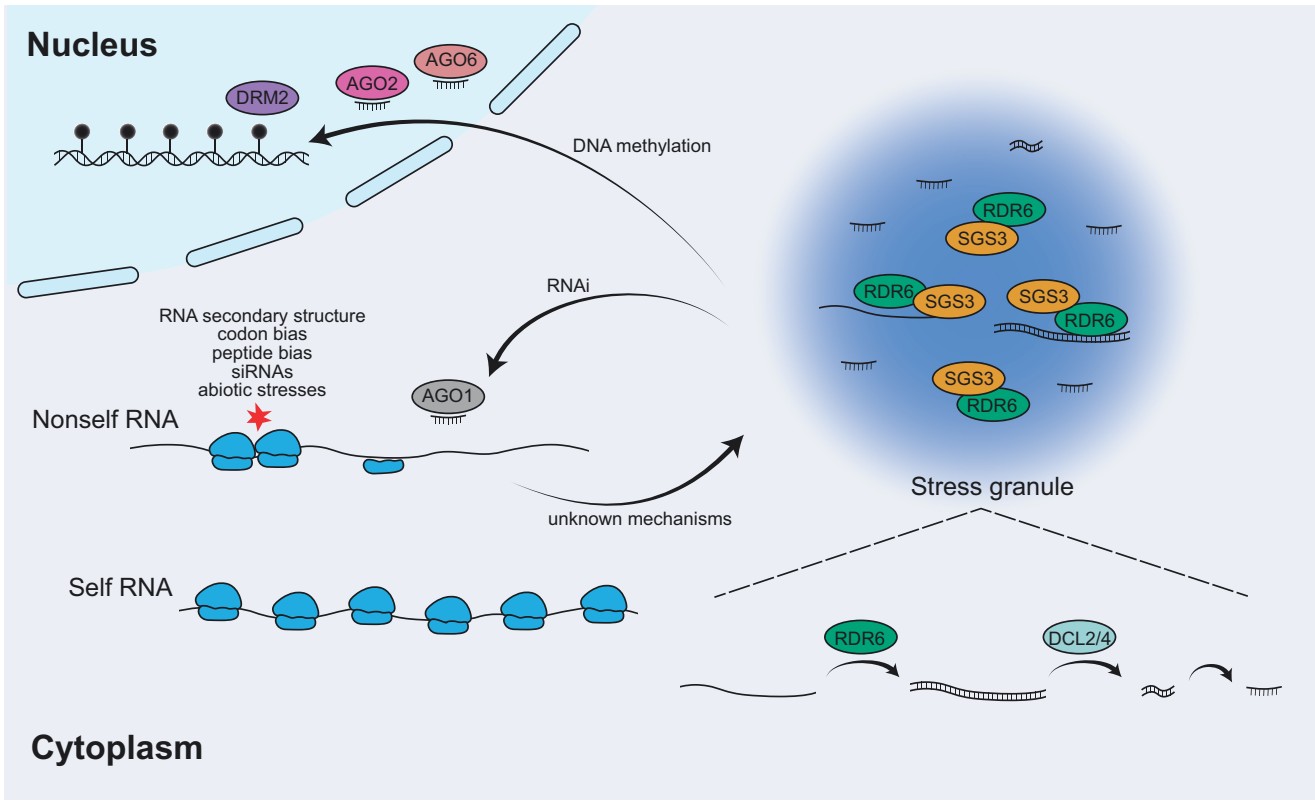

**Figure 1.** RDR6-dependent RNA-directed DNA methylation pathway. Host cells recognise nonself RNAs by distinctive features such as reduced translational efficiency and ribosome stalling caused by various factors. These RNAs are subsequently directed to siRNA bodies through mechanisms that remain unclear. The assembly of siRNA bodies relies on SGS3-driven phase separation, which recruits RDR6 to these sites. The resulting siRNAs then trigger RNA interference (RNAi) and initiate *de novo* DNA methylation. In diagrams, closed circles attached to DNA represent methylation, while ribosomes are shown in blue aligned along RNAs.

Pol IV-mediated or canonical RdDM is largely limited to heterochromatic regions, and is less effective at targeting DNAs that are unmarked by repressive epigenetic modifications. However, an alternative RdDM pathway, often referred to as RDR6-dependent RdDM, expands the silencing potential to new TEs or exogenous sequences, such as transgenes and viral RNAs (Cuerda-Gil & Slotkin, 2016; Fultz et al., 2015; Hung & Slotkin, 2021). This pathway is initiated when RDR6 converts the Pol II-transcribed and invasive RNAs into double-stranded RNAs, which are then processed by DCL2 and DCL4 into 21/22-nt siRNAs. These siRNAs are loaded onto AGO1, AGO2 or AGO6, which can trigger *de novo* DNA methylation at homologous genomic loci (Creasey et al., 2014; Garcia et al., 2012; McCue et al., 2015; Panda et al., 2016; Pontier et al., 2012). While less well-characterised than canonical RdDM, this RDR6-dependent pathway represents a flexible and adaptive silencing system capable of initiating epigenetic silencing at novel targets, even in the absence of pre-existing methylation or heterochrom atic marks (Figure 1).

Since alternative RdDM initiates at the RNA stage, it cannot rely on the epigenetic cues that guide canonical RdDM; instead, its target specificity depends on sequence or structural signatures encoded within the RNA itself. For example, it has been well documented that RDR6 preferentially targets incomplete, truncated or aberrant RNAs (Baeg et al., 2017; Luo & Chen, 2007). One major cause is cleavage by 22-nt miRNAs, which are particularly effective at initiating secondary siRNA production (Cuperus et al., 2010; de Felippes et al., 2017; Iwakawa et al., 2021; Song et al., 2012; Yoshikawa et al., 2013, 2021). Moreover, the so-called two-hit model further suggests that the action of two small RNAs on the same transcript, one of these sites being uncleavable, enhances RDR6 activity, likely by increasing access for double-stranded RNA synthesis (Axtell et al., 2006). In addition to cleavage-based triggers, RNAs with strong secondary structures, such as those derived from inverted repeats, or transcripts formed through overlapping sense and antisense transcription can produce double-stranded regions that mimic RDR6 substrates (Wroblewski et al., 2014; Zhang et al., 2022). Importantly, all of these known pathways depend on pre-existing sequence complementarity, whether through miRNAs, siRNAs, or antisense RNAs, meaning that it serves to act as an adaptive process, rather than an innate mechanism, against foreign genetic elements. In the following section, recent findings around the innate immunity-like recognition of alien RNAs preceding the epigenetic silencing will be discussed.

## 2. Main

### 2.1. Translation-associated RNA cleavage

RNA cleavage is an essential prerequisite for the activity of RDR6, the key initiator of alternative RdDM, and complete and normal transcripts are immune to this process. For example, Creasey et al. suggested that miRNA-mediated RNA cleavage can trigger the production of TE-associated siRNAs, termed epigenetically activated siRNAs (easiRNAs), generated in the DNA methylation-deficient *Arabidopsis* mutants (Creasey et al., 2014). Although several miRNA-independent pathways, such as mRNA splicing (Dalakouras et al., 2019; Kanno et al., 2023; Oberlin et al., 2017), RNA processing (Dadami et al., 2013; Elvira-Matelot et al., 2016)

and RNA surveillance pathways (Martínez de Alba et al., 2015; Moreno et al., 2013; Szádeczky-Kardoss et al., 2018), have been proposed to initiate the RNA cleavage required for alternative RdDM, these mechanisms appear to act only on specific transcripts and do not broadly explain the targeting specificity of alternative RdDM.

Previous studies have identified GC content as a key factor in ensuring robust transgene expression and preventing entry to the alternative RdDM pathway (Brule & Grayhack, 2017; Parret et al., 2016; Sidorenko et al., 2017). This finding is further supported by evidence that TEs tend to use codons that are suboptimal for the host translational machinery, resulting in weaker translational activities compared to endogenous genes (Kim et al., 2021). A similar reduction of translational activity was observed in the transcripts of a retrotransposon, known as *EVADE* (Oberlin et al., 2022). Kim et al. also reported significant differences in GC3 content (GC content at the 3rd nucleotide position of a codon) between genes and TEs across diverse eukaryotic organisms, suggesting that codon usage and the resulting translational efficiency may serve as a universal mechanism for host genomes to discriminate self from non-self RNAs (Kim et al., 2021). Given that viruses, as being foreign to their hosts, typically exhibit codon usage patterns distinct from that of host (Belalov & Lukashev, 2013; Gaunt & Digard, 2022; Jitobaom et al., 2020; Plant & Ye, 2022), it is plausible that translational activity, determined by codon composition, contributes broadly to recognition and silencing of foreign genetic elements.

Translational inefficiency not only reduces protein output but also can trigger RNA quality control mechanisms that result in RNA cleavage, a critical prerequisite essential for RDR6 activity in alternative RdDM. One such mechanism is the no-go RNA decay (NGD) pathway, which targets transcripts where ribosomes stall during translation (Inada & Beckmann, 2024; X. Li et al., 2025; Monaghan et al., 2023; Müller et al., 2025). These stalls can be caused by strong secondary structures, rare codons, disome formation (ribosome collisions) or other features that impede ribosome progression. In response, the NGD machinery induces endonucleolytic cleavage near the stall site, generating RNA fragments that can potentially serve as substrates for RDR6. Consistent with this notion, Oberlin et al. was able to detect cleaved RNA ends at where ribosome is stalled in *EVADE* RNA (Oberlin et al., 2022), and Kim et al. found signals for frequent RNA cleavage and stacked ribosomes enriched in TEs of *Arabidopsis* (Kim et al., 2021). This positions NGD as a potential contributor to the initiation of siRNA production and silencing via alternative RdDM.

Emerging evidence suggests that NGD operates in plants, particularly in controlling non-native RNAs like that of viruses. For example, recent studies have shown that NGD factors, such as PELOTA1 and HBS1, play a direct role in suppressing viral infection in plants (Ge et al., 2023; Pouclet et al., 2023). In *Arabidopsis*, these proteins recognise conserved A-rich motifs ($G_{1-2} A_{6-7}$) in potyvirus genomes and trigger endonucleolytic cleavage of viral RNAs, limiting their accumulation. This mirrors the observation that TEs are typically AT-rich, a feature associated with suboptimal translational efficiency (Boissinot, 2022; Kim et al., 2021). Moreover, a recent study suggests that aberrant RNAs arise from ribosome stalling prior to the full establishment of transgene silencing (Kramer et al., 2025). These findings together highlight NGD as a targeted antiviral mechanism, linking translational surveillance to gene silencing and reinforcing the idea that inefficient translation may mark transcripts as non-self or aberrant.

The siRNA pathway in plants offers a strategic advantage over simple RNA degradation by acting more like an adaptive immune system. It enables sequence-specific recognition of foreign or aberrant RNAs and converts these into heritable silencing signals through RdDM. Unlike transient RNA decay, siRNA-mediated silencing can be amplified, systemic and long-lasting, ensuring robust and persistent defence. While there is rich, albeit circumstantial, evidence suggesting that NGD may act as a trigger for the initial RNA cleavage required to initiate the alternative RdDM pathway, direct empirical evidence linking NGD to siRNA biogenesis remains to be established. Once siRNAs are produced and DNA methylation is established at target loci, the canonical RdDM machinery can further strengthen and maintain the silenced state.

## 2.2. Localisation to siRNA bodies

RDR6 has been shown to localise to stress granules (SGs), cytoplasmic compartments formed particularly under environmental stress conditions (Kim et al., 2021; Kumakura et al., 2009; Moreno et al., 2013; Tan et al., 2023; Wen et al., 2024). SGs themselves are dynamic cytoplasmic structures that temporarily store and sort untranslated mRNAs during stress, helping cells modulate gene expression and maintain homeostasis (Ren et al., 2025; Zhao & Li, 2025). In *Arabidopsis*, RDR6-containing SGs, also known as siRNA bodies, appear to form through liquid–liquid phase separation (LLPS) mediated by SGS3. Deletion of the LLPS domain in SGS3 disrupts its ability to localise to these granules, which in turn prevents the recruitment of RDR6 and impairs siRNA biogenesis (Kim et al., 2021; Tan et al., 2023) (Figure 2).

The formation of siRNA bodies adds an additional layer of specificity and selectivity in the system as not all transcripts are equally guided to these compartments. Studies have shown that RNAs with reduced translational activity are more strongly enriched in SGs (Helton et al., 2025; Khong et al., 2017; Khong & Parker, 2018; Kim et al., 2021; Matheny et al., 2019, 2021). This selective enrichment of poorly translating RNAs in SGs appears to be evolutionarily conserved as it has been observed across diverse eukaryotes, including yeast, humans and plants. Although the precise biochemical mechanisms guiding ribosome-depleted RNAs to SGs remain unclear, these findings support the model that translational inefficiency is a key determinant in directing transcripts to the alternative RdDM pathway by funnelling them into cellular compartments where essential silencing components are localised.

In addition to reduced translation, N6-methyladenosine ($m^6A$) RNA modification has been shown to promote the localisation of specific transcripts to SGs (Di Timoteo et al., 2024; Fan et al., 2023; Li et al., 2024; Song et al., 2024; Wu et al., 2024). A recent study in *Arabidopsis* demonstrated that $m^6A$-marked RNAs are preferentially enriched in these compartments, and this localisation depends on $m^6A$ reader proteins that facilitate LLPS (Fan et al., 2023; Wu et al., 2024). This suggests that $m^6A$ may serve as a licensing mark, guiding transcripts into the silencing pathway. While a direct role of $m^6A$ in promoting siRNA biogenesis remains to be confirmed, these observations support a model in which $m^6A$ facilitates the selective targeting of transcripts for silencing through compartmentalisation.

TE RNAs in plants are marked with high levels of $m^6A$, reflecting a strategic layer of post-transcriptional regulation by the host. For example, in *Arabidopsis*, the retrotransposon *Onsen* is methylated by $m^6A$ upon heat activation, and this methylation restricts its activity by retaining its RNA in SGs (Fan et al., 2023). Similarly, TEs from other eukaryotes, such as LINE-1 elements in humans,

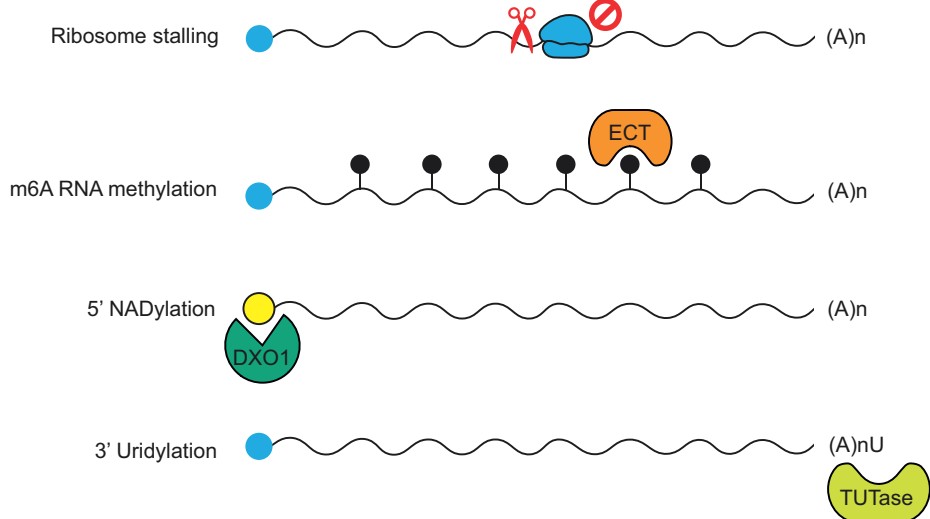

**Figure 2.** RNA pathways initiating siRNA biogenesis. RNA cleavage is a critical prerequisite for entry into the siRNA pathway and can be initiated through multiple mechanisms. Ribosome stalling, often caused by suboptimal codons, can induce RNA cleavage and promote RNA localisation to siRNA bodies. m6A-modified RNAs are commonly linked to RNA destabilisation and stress granule localisation, yet the contribution of m6A-binding ECT (EVOLUTIONARILY CONSERVED C-TERMINAL DOMAIN) family proteins to this process remains unclear. Additional RNA modifications, including 5′ NADylation and 3′ uridylation, catalysed by DXO1 (DECAPPING AND EXORIBONUCLEASE PROTEIN 1) and TUTases (TERMINAL URIDYLYLTRANSFERASES), respectively, are closely associated with RNA degradation and siRNA biogenesis. Blue circle, m7G cap; black circle, m6A RNA methylation; yellow circle, NAD+ cap.

are also modified by $m^6$A (Barter & Cho, 2025; Hwang et al., 2021; Wei et al., 2022; Xiong et al., 2021). Beyond TEs, many studies have identified $m^6$A RNA methylation on viral RNAs as well (Y. Li et al., 2025; Manners et al., 2019; Zhang et al., 2023), suggesting that this epitranscriptomic mark broadly functions to label non-native RNAs. However, because the same RNA modification also regulates endogenous mRNAs, $m^6$A-mediated differentiation may rely on additional molecular signatures. For instance, transposon RNAs are often methylated at multiple sites throughout their sequence, whereas typical mRNAs usually exhibit a single methylation peak near stop codons (Fan et al., 2023). Such differences in the number and position of $m^6$A marks may help distinguish non-native RNAs. Altogether, $m^6$A might act as a host-encoded marker to selectively flag foreign RNAs, guiding them into silencing pathways and reinforcing genomic defence at the RNA level.

The localisation of RNAs to siRNA bodies represents a crucial selective step in siRNA production. This selective guidance relies on key features including reduced translational activity and $m^6$A RNA methylation, which together help funnel target RNAs into the alternative RdDM pathway. Despite growing evidence supporting the importance of SG formation in RNA silencing, major knowledge gaps still remain. Notably, the precise molecular mechanisms by which $m^6$A modifications and translational repression coordinate to drive RNA partitioning into SGs are still unclear. Additionally, whether similar principles apply universally across different organisms and RNA types warrant further investigation. Addressing these questions will be critical to fully understanding how cells maintain genomic stability through epitranscriptomic and translational control.

### 2.3. RNA decay versus RNA silencing

So far, we have examined the functional impacts of translational inactivation on the initiation of siRNA biogenesis, focusing on RNA cleavage and localisation to siRNA bodies, both of which

confer RNA specificity to the silencing pathway. We also discussed NGD as a potential mediator of translation-associated cleavage of non-native RNAs. However, as a host RNA quality control mechanism, NGD primarily promotes RNA degradation rather than siRNA production (Deragon & Merret, 2025; Szádeczky-Kardoss et al., 2018; Wu et al., 2023; You et al., 2019; Zhang et al., 2015). In this section, we review current knowledge on the interplay between RNA decay and RNA silencing in plants.

The relationship between RNA decay and siRNA biogenesis in plants is often antagonistic, with RNA decay acting as a primary RNA quality control mechanism that can override RNAi (Christie et al., 2011; Kim, 2023). Both pathways target aberrant or excessive RNAs in processing (P) bodies and siRNA bodies, respectively, but RNA decay typically acts first to prevent the activation of silencing responses. This prioritisation is crucial because RNA silencing, once triggered, can lead to the unintended silencing of functional endogenous genes. Thus, RNA decay serves to suppress inappropriate RNA silencing activity by rapidly degrading faulty RNAs before they can be recognised by the siRNA machinery. For instance, studies in *Arabidopsis* have shown that mutants deficient in DECAPPING 2 (DCP2) or EXORIBONUCLEASE 4 (XRN4), the P body components, accumulate aberrant RNAs that are subsequently processed into siRNAs, triggering gene silencing (Elvira-Matelot et al., 2016; Gregory et al., 2008; Gy et al., 2007; Li & Wang, 2018; Martínez de Alba et al., 2015; Sorenson et al., 2018; Souret et al., 2004; Thran et al., 2012). These findings illustrate how RNA decay pathways act to suppress RNA silencing under normal conditions, ensuring that RNA silencing remains a specialised response reserved for transgenes, viruses or highly abnormal transcripts, rather than ordinary endogenous mRNAs.

Although structurally complete and intact, RNAs modified at the 5′ and 3′ ends, such as nicotinamide adenine dinucleotide ($NAD^+$) capping and 3′ uridylation, can be directed into RNA stability or silencing pathways. At the 5′ end, some RNAs bear non-canonical caps like $NAD^+$ instead of the typical $m^7$G cap. In *Arabidopsis*, $NAD^+$-capped RNAs are recognised as aberrant

and are preferentially targeted for degradation, limiting their potential to engage in RNA silencing (Carpentier et al., 2025; Kwasnik et al., 2019; Wang et al., 2019). Conversely, other studies suggest that NAD$^+$-capped RNAs can be recruited to RDR6 and processed to siRNAs (Pan et al., 2020; Yu et al., 2021). At the 3′ end, uridylation, the addition of uridines by terminal uridylyl transferases (TUTases), can similarly mark RNAs for degradation or processing into small RNAs. In plants, uridylation of viral RNAs or transgene-derived transcripts promotes their degradation and enhances silencing (de Almeida et al., 2018; Joly et al., 2023; Scheer et al., 2021; Wang et al., 2022). These modifications act as molecular tags that distinguish normal RNAs from those targeted for silencing, thereby ensuring precise regulation of gene expression and defence. However, the mechanisms by which specific RNA modifications are selected to channel RNAs into either degradation or silencing pathways remain incompletely understood, and further research is needed to elucidate the determinants governing these RNA fates.

## 3. Conclusion and future perspectives

Despite significant advances in our understanding of RNA silencing in plants, critical questions remain regarding the earliest steps of pathway initiation. One major gap concerns the molecular signals that guide the recruitment of RDR6 to aberrant RNAs. Although translation stalling and RNA quality control mechanisms have been implicated, the precise criteria by which transcripts are selected for RDR6-mediated dsRNA synthesis remain unclear. The RNA-binding protein SGS3, which interacts with RDR6 and contributes to siRNA processing, is a potential determinant of target selection, but its exact role in this context is not fully understood. Insights from yeast offer additional clues. In yeast, NGD has been shown to generate 3′ cleavage fragments with 5′ hydroxyl (5′OH) ends, in contrast to the 5′ phosphorylated ends typically produced by other endoribonucleases (Navickas et al., 2020). Notably, RNA degradation pathways that follow ribosome-associated RNA cleavage generally require 5′ phosphate groups, and the tRNA ligase Trl1 can phosphorylate 5′OH termini, thereby licensing them for decay. This raises the intriguing possibility that a similar mechanism might exist in plants to regulate the fate of NGD-generated fragments. Mapping 5′OH-RNAs in plants and identifying host enzymes analogous to Trl1 would be a compelling direction for future research. Additionally, the localisation of NGD-related events within siRNA bodies further suggests a spatial link between RNA decay and RNA silencing, warranting deeper investigation into their potential coordination.

RNA silencing is highly responsive to environmental cues, including pathogen attack, drought, UV exposure and nitrogen deprivation (Kallemi et al., 2024; Li et al., 2023; Lopez-Gomollon & Baulcombe, 2022; Pothof et al., 2009; Westwood et al., 2013; Wu et al., 2020). However, the molecular basis by which such stimuli selectively initiate silencing remains unresolved. Future studies should apply time-resolved transcriptomics and small RNA-seq after environmental stimuli to identify early-responding genes and siRNA populations. Once initiated, RNA silencing can amplify itself via secondary siRNA production, yet the mechanisms that constrain or fine-tune this amplification are not well characterised. Experiments that titrate dsRNA or trigger RNA levels using synthetic constructs or inducible promoters could be used to establish these thresholds *in vivo*.

Understanding how plants detect and epigenetically silence foreign genetic elements is crucial for unravelling the complexities of genome defence and stability. While much progress has been made in identifying the role of siRNAs in maintaining silencing, the early events that initiate this response are only beginning to be elucidated. Continued investigation into these upstream processes will not only deepen our fundamental knowledge of plant immunity but also inform the development of more stable and predictable transgenic technologies in agriculture and biotechnology.

**Open peer review.** To view the open peer review materials for this article, please visit http://doi.org/10.1017/qpb.2025.10029.

**Competing interest.** The authors declare none.

**Author contributions.** SM drafted the manuscript, and YJK and JC edited and revised the manuscript.

**Funding statement.** This work was supported by the National Research Foundation of Korea (RS-2024-00336161 to YJK) and UKRI-BBSRC (UKRI1915 to JC).

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
