## [Reviewer Report]

The manuscript entitled “Small RNA, big defense: Early epigenetic responses to genetic invasion” reviewed recent findings on how plant cells recognize foreign nucleic acids and initiate epigenetic silencing. The authors discussed these findings from three perspectives: translation-associated RNA cleavage, localization to siRNA bodies, and RNA decay versus RNA silencing. By integrating these emerging mechanisms with established pathways, such as RNA-directed DNA methylation (RdDM) and non-conanical RdDM pathways, primarily executors for silencing foreign nucleic acids, the manuscript advances our understanding of how plant cells distinguish foreign nucleic acids from their own. The authors, who themselves contributed significantly to this field, offer reasonable and insightful interpretation and perspective. Overall, the manuscript is well-written, concise, and accurate. I really enjoyed reading it.

Minor concerns:

1, Both figures are somewhat oversimplified. First, key symbols (for example, ribosomes and DNA methylation in Figure 1) should be clearly defined in the figures or figure legends. Second, figure 1 focuses on ribosome stalling and siRNA body. It would be better to add some details on why and how non-self RNAs are experiencing ribosome stalling and how these non-self RNAs are being processed to be delivered into the siRNA body. One solution is to combine figures 1 and 2 into a large figure with more mechanisms included.

2, On page 7, line 201, the authors stated that “m6A acts as a host-encoded marker to selectively flag foreign RNAs, guiding them into silencing pathways and reinforcing genomic defense at the RNA level.” I think the statement might not be accurate because the majority of the plant mRNAs have m6A modifications. Therefore, m6A modifications on foreign RNAs may not be necessary to label or distinguish non-native RNAs. There must be some other mechanisms together with m6A modifications to label non-native RNAs. I recommend the authors discuss the complexity and possibility of different mechanisms.

3, On page 7, line 214-237, when discussing RNA decay pathways act to suppress RNA silencing under normal conditions, the authors likely treat NGD and RNA decay as the same mechanism. I think the authors may need to add some details to distinguish these two RNA decay pathways. For example, normal RNA decay occurred in processing bodies. DCP2 and XRN4 are key components of processing bodies, and their deficiency leads to aberrant processing into siRNA in siRNA bodies. NGD, which targets transcripts with stalled ribosomes, may actively contribute to silencing initiation by generating cleavage RNA fragments. I hope the authors can explain more details to clarify the differences between these mechanisms.

---

## [Reviewer Report]

This review focuses on the epigenetic defense mechanisms against “genetic invasion” from exogenous genetic elements (e.g., viruses, bacteria, transgenes) and endogenous threats (e.g., transposable elements, TEs) in plants. It emphasizes the central role of small interfering RNAs (siRNAs) and RNA-directed DNA methylation (RdDM) in these processes, while dissecting the understudied early steps of epigenetic silencing initiation. This manuscript was written well and covered major recent progresses in this area. Still, it needs some minor improvements to be accepted. My comments are:

1. Line 143, in this part on NGD, the most recent paper on aberrant RNA and transgene silencing (Kramer, et al., 2025) should be covered and cited.

2. Line 153, the disome from ribosome collision was reportedly related to NGD as well. The authors may discuss the role of the architecture of ribosomes and mRNA in NGD and RQC that result in aberrant RNA and siRNA biogenesis.

3. Line 219, recent papers on siRNA generated from insufficient mRNA degradation should also be cited, such as Zhang, et al. 2015, Science and You, et al. 2019, Nat. Commun.

4. Line 274, the Nature paper (Wu, et al. 2020) on 22-nt siRNA and nitrogen-depletion discussed the relationship between environmental cues and translational repression. Though RdDM-mediated gene silencing was not mentioned in this paper, the authors may also discussed environment-induced siRNAs playing roles in DNA methylation.